# Early recovery of proteasome activity in cells pulse-treated with proteasome inhibitors is independent of DDI2

**Ibtisam Ibtisam, Alexei F Kisselev***

Department of Drug Discovery and Development, Harrison College of Pharmacy, Auburn University, Auburn, United States

**Abstract** Rapid recovery of proteasome activity may contribute to intrinsic and acquired resistance to FDA-approved proteasome inhibitors. Previous studies have demonstrated that the expression of proteasome genes in cells treated with sub-lethal concentrations of proteasome inhibitors is upregulated by the transcription factor Nrf1 (NFE2L1), which is activated by a DDI2 protease. Here, we demonstrate that the recovery of proteasome activity is DDI2-independent and occurs before transcription of proteasomal genes is upregulated but requires protein translation. Thus, mammalian cells possess an additional DDI2 and transcription-independent pathway for the rapid recovery of proteasome activity after proteasome inhibition.

## eLife assessment

The study presents **important** findings regarding a transcription-independent component of the early recovery of proteasome activity from a short pulse of proteasome inhibitor treatment, which has not been appreciated before and which is independent of the DDI2-NRF2 axis. While the evidence is in principle **solid**, with recapitulation in several cell line models, the proposed alternative underlying mechanism, namely regulation at the level of proteasome assembly, lacks experimental support, and at this point remain speculative.

**\*For correspondence:**
AFK0006@auburn.edu

## Introduction

The ubiquitin-proteasome system is the primary protein quality control pathway in every eukaryotic cell. By degrading numerous regulatory proteins, this pathway also plays a pivotal role in regulating many cellular functions such as cell cycle and gene expression. Malignant cells are more dependent on proteasome function than non-transformed cells because they divide rapidly and produce abnormal proteins at a higher rate than normal cells (*Deshaies, 2014*; *Kisselev et al., 2012*). Proteasome inhibitors (PIs) bortezomib (Btz), carfilzomib (Cfz), and ixazomib are approved for the treatment of multiple myeloma (MM). Btz is also approved for the treatment of mantle cell lymphoma (MCL). MM cells are exquisitely sensitive to PIs because the production of immunoglobulins by these malignant plasma cells places an enormous load on the proteasome and other components of the protein quality control machinery (*Cascio et al., 2008*; *Bianchi et al., 2009*; *Cenci et al., 2011*; *Shabaneh et al., 2013*).

Clinically, Btz and Cfz are administered once or twice weekly as a subcutaneous (Btz) or intravenous bolus. They cause rapid inhibition of proteasome activity in the patients' blood but are metabolized rapidly (*Wang et al., 2021*). Within an hour after the administration, PIs concentrations in the blood drop below the levels needed to kill tumor cells *in vitro* (*Hamilton et al., 2005*; *Moreau et al., 2011*). Although Btz has a very slow off-rate and Cfz is an irreversible inhibitor, proteasome activity recovers within 24 hr (*Shabaneh et al., 2013*; *Hamilton et al., 2005*; *O'Connor et al., 2009*; *Weyburne*

*et al., 2017*). This activity recovery may explain discrepancies between robust activity against cell lines derived from various cancers, continuously treated with Btz (http://www.carcerrxgene.org/) (*Shabaneh et al., 2013*), and a lack of clinical efficacy except in MM and MCL. In addition, recovery of activity has recently been implicated in PI resistance in MM (*Op et al., 2022*).

In cells treated with PIs, a transcription factor Nrf1 (also known as TCF11, encoded by the *NFE2L1* gene) upregulates the transcription of genes encoding all proteasome subunits (*Steffen et al., 2010*; *Radhakrishnan et al., 2010*). When the proteasome is fully functional, Nrf1 is constitutively degraded in a ubiquitin-dependent manner (*Steffen et al., 2010*). When the proteasome is partially inhibited, the ubiquitylated Nrf1 is recognized by DDI2 (DNA-Damage-Inducible I Homolog 2), a ubiquitin-dependent aspartic protease that activates Nrf1 by a site-specific cleavage (*Koizumi et al., 2016*; *Lehrbach and Ruvkun, 2016*). Although knockdown of DDI2 blocks the PI-induced transcription of proteasome genes (*Koizumi et al., 2016*; *Lehrbach and Ruvkun, 2016*), initial studies implicating DDI2 in the activation of Nrf1 did not determine whether DDI2/Nrf1-dependent transcription leads to the recovery of activity after clinically relevant pulse treatment with PIs. In this work, we asked whether DDI2 is involved in activity recovery after such treatment. Unexpectedly, we found that proteasome activity recovered in the absence of DDI2, and activity recovery preceded the upregulation of proteasome genes. This data demonstrates the existence of a novel, DDI2-independent pathway for the recovery of proteasome activity in PI-treated cells.

## Results

To analyze DDI2 involvement in the recovery of proteasome activity after treatment with PIs, we used commercially available clones of HAP1 cells, in which DDI2 was knocked out by CRISPR, and a clone with an unaltered DDI2, which we will refer to as a wild type (wt, *Figure 1a*). We analyzed three different clones that were generated by using two different gRNAs (Key Resources Table). We treated cells for 1 hr with a range of concentrations of Cfz and Btz and then cultured them in drug-free media (*Figure 1b*). We measured inhibition of the proteasome's β5 site, which is the prime target of Cfz and Btz (*Kisselev et al., 2012*), immediately after the 1 hr treatment, and 12 or 24 hr thereafter (*Figure 1c*), which is when recovery plateaued (not shown). In a parallel experiment, we used a Cell-Titer-Glo assay, which measures intracellular ATP levels, to determine cell viability 12 and 24 hr after treatments (*Figure 1c*). Initial inhibition of proteasome was observed at sub-lethal concentrations, and proteasome activity recovered in cells treated with such concentrations. Surprisingly, no differences in the recovery between wt and DDI2-KO clones were observed (*Figure 1c*). Deletion of DDI2 did not affect recovery, despite inhibition of Btz-induced proteolytic activation of Nrf1 (*Figure 1d* and *Figure 1—figure supplement 1*). These findings confirm that DDI2 activates Nrf1, but indicate that it is not involved in the recovery of proteasome activity in Btz and Cfz-treated HAP1 and DDI2 KO cells.

Next, we knocked down DDI2 by two different highly efficient siRNAs in two PI-sensitive triple-negative breast cancer cell lines, SUM149 and MDA-MB-231 (*Figure 1e*). Proteasome activity in these cells and their sensitivity to PIs were similar to HAP1 cells (*Figure 1—figure supplement 2*). The knockdown did not significantly affect the recovery of proteasome activity in cells treated with 100 nM Btz (*Figure 1f*). Finally, we found that inactivation of DDI2 by the D252N mutation of the catalytic aspartic acid residue (*Koizumi et al., 2016*) did not block the recovery of activity after pulse treatment of HCT-116 cells with Btz and Cfz (*Figure 1g*). Thus, the recovery of proteasome activity after pulse treatment with sub-toxic concentrations of PIs is DDI2-independent.

If inhibitor-induced transcription of proteasome genes is responsible for the recovery of proteasome activity, the upregulation of proteasome gene expression should precede the activity recovery. However, we found that the recovery of activity started immediately after 1 hr pulse treatment and approached a plateau after 8 hr (*Figure 2a*), but the first significant increase in the expression of proteasomal mRNAs occurred only 8 hr after the removal of the inhibitor (*Figure 2b*). These results suggest that the early recovery of the proteasome activity is not a transcriptional response.

Ruling out transcriptional response does not rule out the production of new proteasomes because protein synthesis can be regulated at the translational level. To determine whether the activity recovery involves the biosynthesis of new proteasomes, we studied the effects of cycloheximide (CHX), an inhibitor of protein biosynthesis, on the recovery. Except for the first hour, the recovery was completely blocked by CHX, independent of DDI2 expression status (*Figure 3a*). Thus, the recovery of proteasome activity involves protein synthesis.

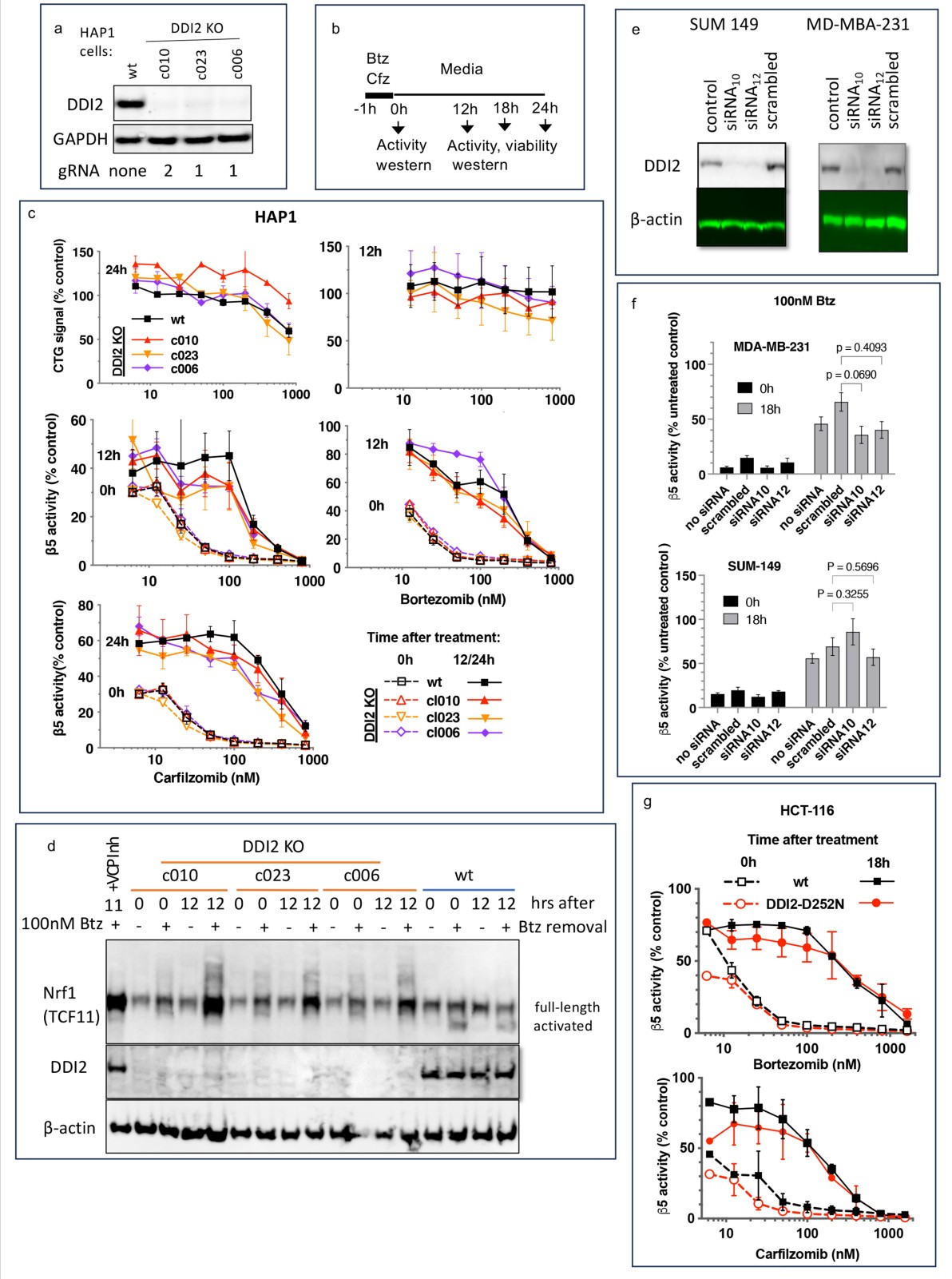

**Figure 1.** Recovery of proteasome activity is DDI2 independent. (**a**) Expression of DDI2 in the CRISPR-generated clones of HAP1 cells used in this work was analyzed by western blot. (**b**) The experimental setup used in this study. Cells were pulse treated with bortezomib (Btz) or carfilzomib (Cfz) for 1 hr, then cultured in drug-free media for times indicated and analyzed as described. (**c**) The viability of wt- and DDI2 KO clones of HAP1 cells was measured using CellTiter-Glo, and the inhibition of β5 sites was measured with the Proteasome-Glo assay at times indicated; n=2–5. (**d**) Knockout of DDI2 inhibits

*Figure 1 continued on next page*

*Figure 1 continued*

the Nrf1 processing. Western blots of Btz-treated HAP1 cells. The sample in the first lane is wt cells treated with VCP/p97 inhibitor CB-5083 immediately after removal of Btz. VCP inhibitors blocks Nrf1 processing (*Radhakrishnan et al., 2014*; *Sha and Goldberg, 2014*; *Anderson et al., 2015*). (**e**) MDA-MB-231 and SUM149 cells were analyzed by western blot 72 hr after transfection with DDI2 siRNAs (**f**) Theβ5 activity in siRNA-transfected SUM149 and MDA-MB-231 was measured using Suc-LLVY-AMC immediately and 18 hr after treatment with 100 nM Btz; n=3. (**g**) β5 activity was measured in HCT-116 cells with the Proteasome-Glo assay immediately and 18 hr after treatment with PIs; n=2.

The online version of this article includes the following source data and figure supplement(s) for figure 1:

**Source data 1.** PDF file containing *Figure 1a* and original full-size western blot membranes (anti-DDI2, anti-GAPDH) with molecular weight markers.

**Source data 2.** Excel file containing data for *Figure 1c*.

**Source data 3.** PDF file containing *Figure 1d* and original full-size western blot membranes (anti-Nrf1, anti-DDI2, anti-β-actin) with molecular weight markers.

**Source data 4.** PDF file containing *Figure 1e* and full-size western blot membranes (anti-DDI2, anti-β-actin).

**Source data 5.** Excel file containing data and statistical analysis for *Figure 1f*.

**Source data 6.** Excel file containing data for *Figure 1g*.

**Figure supplement 1.** The proteasome activity of the samples used in *Figure 1d* was measured with Suc-LLVY-AMC; n=9.

**Figure supplement 1—source data 1.** Excel file containing data and statistical analysis.

**Figure supplement 2.** Comparison of proteasome activity and proteasome inhibitor (PI) sensitivity between HAP1, MDA-MB-231, and SUM149 cells.

**Figure supplement 2—source data 1.** Excel file containing data for both panels.

Activation of proteasomal mRNA translation could explain transcription-independent production of new proteasomes if a significant fraction of proteasomal mRNAs is untranslated in the absence of PI treatment. We used polysome profiling to determine the distribution of proteasomal mRNA between translated and untranslated fractions. We found that 90% of proteasomal mRNAs are ribosome or polysome bound in untreated cells (*Figure 3b*), and treatment with inhibitors did not increase this fraction (*Figure 3—figure supplement 1*). This result agrees with a published result that the amount of proteasome mRNA in the polysomal fraction does not increase when proteasome is inhibited in MM1.S cells (*Wiita et al., 2013*). Thus, the biosynthesis of active new proteasomes immediately after treatment with sub-lethal concentrations of PIs appears to occur without upregulation of translation of mRNAs encoding proteasome subunits.

## Discussion

The most important conclusion of this work is that, in addition to Nrf1/DDI2 pathway, mammalian cells possess at least one additional pathway to restore proteasome activity after treatment with PIs, and this DDI2-independent pathway is responsible for the rapid synthesis of new proteasomes immediately after treatment with PI. While this study was underway, two other laboratories found that knockout of DDI2 reduced recovery of proteasome activity in multiple myeloma and NIH-3T3 cells pulse-treated with PIs by ~30% (*Chen et al., 2022*; *Northrop et al., 2020*). Similarly, Nrf1 knockdown did not completely block the recovery of proteasome activity in mouse embryonal fibroblasts (*Radhakrishnan et al., 2010*). The clinical impact of our study and these studies in the literature is somewhat limited because we all conducted a single pulse treatment and did not explore whether Nrf1/DDI2 plays a more prominent role in the recovery of proteasome activity after repeated treatment with PIs. These limitations, however, do not question the existence of the DDI2-independent recovery pathway.

Our findings necessitate reconsidering the role of the DDI2/Nrf1 pathway in basal and inhibitor-induced proteasome expression. Previous studies have also reported that DDI2/Nrf1 contributes to the maintenance of basal levels of proteasomes (*Chen et al., 2022*; *Siva et al., 2020*; *Waku et al., 2020*) and that Nrf1 is essential for the basal proteasome expression in the brain (*Lee et al., 2011*), liver (*Lee et al., 2013*), and retina *Wang et al., 2023*; yet, in our experiments, the effects of DDI2 KO on the basal proteasome activity was not significant (*Figure 1—figure supplement 1*). These differences may reflect that heavy secretory MM cells, embryonic cells, and certain specific tissues require higher levels of proteasome activity and use the DDI2/Nrf1 pathway to supplement other pathways responsible for proteasome expression (*Motosugi and Murata, 2019*). Other studies demonstrated

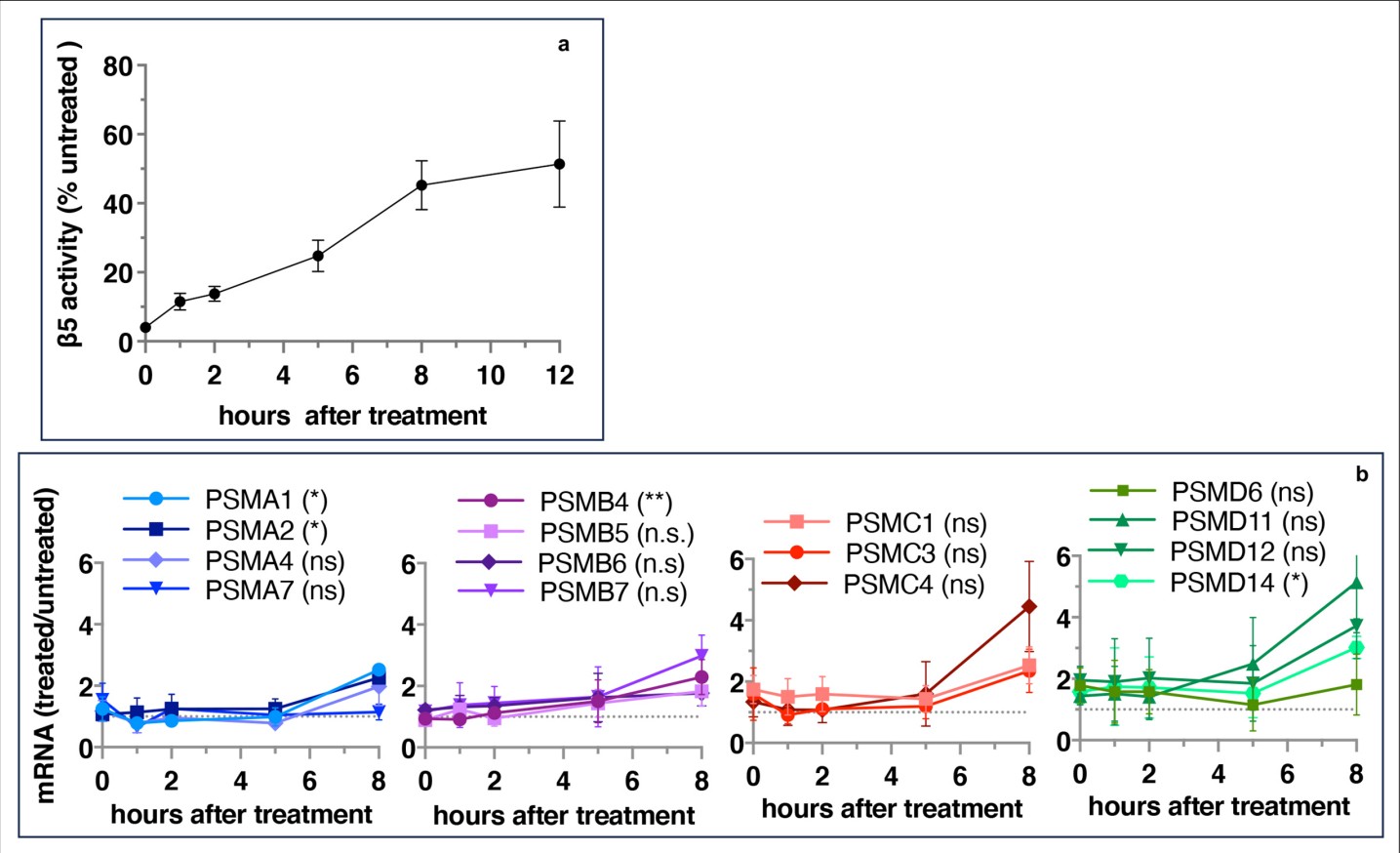

**Figure 2.** Proteasome activity recovers before upregulation of proteasome gene expression. Wt-HAP1 cells were pulse-treated with bortezomib (Btz) (100 nM), cultured in a drug-free medium, and analyzed at indicated times. (**a**) β5 activity was measured using Proteasome-Glo and normalized first to CellTiter-Glo viability data and then to proteasome activity in the mock-treated samples; n=2–5. (**b**) In a parallel experiment, the mRNA was isolated, and the expression of proteasome genes was quantified using quantitative RT-PCR; n=3. Results of the t-test at 8 hr are in parenthesis.

The online version of this article includes the following source data for figure 2:

**Source data 1.** Prism file containing data and statistical analysis for both panels.

the importance of Nrf1-dependent proteasome expression during cardiac regeneration (*Cui et al., 2021*) and thermogenic adaptation of the brown fat (*Bartelt et al., 2018*).

Several studies found that the knockout of DDI2 sensitizes cells to proteasome inhibitors (*Weyburne et al., 2017*; *Op et al., 2022*; *Chen et al., 2022*; *Northrop et al., 2020*; *Dirac-Svejstrup et al., 2020*). This was further interpreted as supportive of a role for DDI2-dependent recovery in the de-sensitization of cells to PI-induced apoptosis. Although we confirmed this observation in HAP1 cells (not shown), the present findings raise a possibility that DDI2 desensitizes cells to PI by a different mechanism. Activation of non-proteasomal Nrf1-dependent oxidative stress response genes (*Ribeiro et al., 2022*; *Kim et al., 2016*) may help overcome the deleterious consequences of PI-induced overproduction of reactive oxygen species (ROS) (*Lipchick et al., 2016*). Alternatively, the ability of DDI2 to bind and participate in the degradation of large ubiquitin conjugates (*Dirac-Svejstrup et al., 2020*; *Collins et al., 2022*) may help alleviate the stress associated with proteasome inhibition. DDI2 and proteasome are involved in DNA repair (*Kottemann et al., 2018*; *Krogan et al., 2004*; *Chen et al., 2010*; *Groisman et al., 2006*; *Aliyaskarova et al., 2023*), and impairment of the proteasome in the absence of DDI2 can lead to excessive spontaneous DNA damage, even without DNA-damaging agents. Finally, the proteolytic activation of another yet-to-be-identified DDI2 substrate cannot be ruled out. In summary, our study provides strong evidence for a novel pathway responsible for the recovery of proteasome activity in inhibitor-treated cells. It should stimulate research on additional biological roles of DDI2, which can explain the embryonic lethality of DDI2 deletion (*Siva et al., 2020*) and DDI2's role in tumorigenesis (*Lei et al., 2023*).

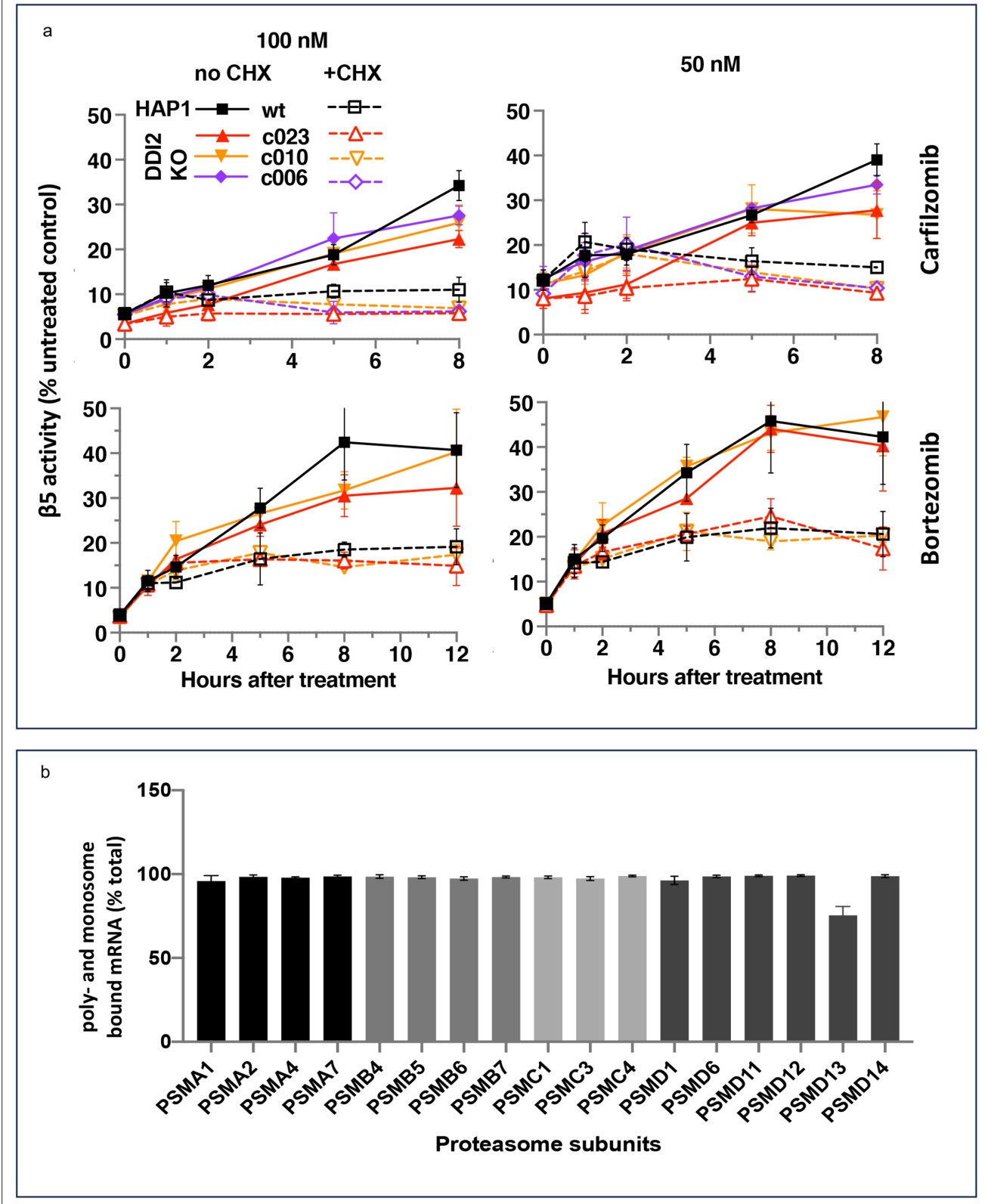

**Figure 3.** The recovery of proteasome activity requires protein synthesis. (**a**) Wt-HAP1 and DDI2 KO cells were treated for 1 hr at indicated concentrations of bortezomib (Btz) and carfilzomib (Cfz) and then cultured in a drug-free media in the absence (solid lines) or presence (dashed lines) of cycloheximide (CHX). The β5 activity was measured using Proteasome-Glo and normalized first to cell viability, which was determined in a parallel

*Figure 3 continued on next page*

*Figure 3 continued*

experiment using CellTiter-Glo, and then to untreated controls; n=3–4. (**b**) All proteasome mRNAs are actively translated. mRNA isolated from untreated wt-HAP1 cells were analyzed by polysome profiling. The combined mRNAs in the 80 S and polysomal fractions as a % of the total is shown; n=2.

The online version of this article includes the following source data and figure supplement(s) for figure 3:

**Source data 1.** Prism file containing data and statistical analysis for *Figure 3a*.

**Source data 2.** Excel file containing data for *Figure 3b*.

**Figure supplement 1.** Translation of catalytic subunits is not altered after treatment with inhibitors.

**Figure supplement 1—source data 1.** Excel file containing data.

## Ideas and speculations

We want to propose a model explaining the upregulated biogenesis of proteasomes without an increase in the efficiency of proteasomal mRNA translation. To gain activity, the catalytic subunits must assemble into mature particles in a complex process involving multiple dedicated chaperones (*Rousseau and Bertolotti, 2018*; *Budenholzer et al., 2017*). The efficiency of nascent subunits

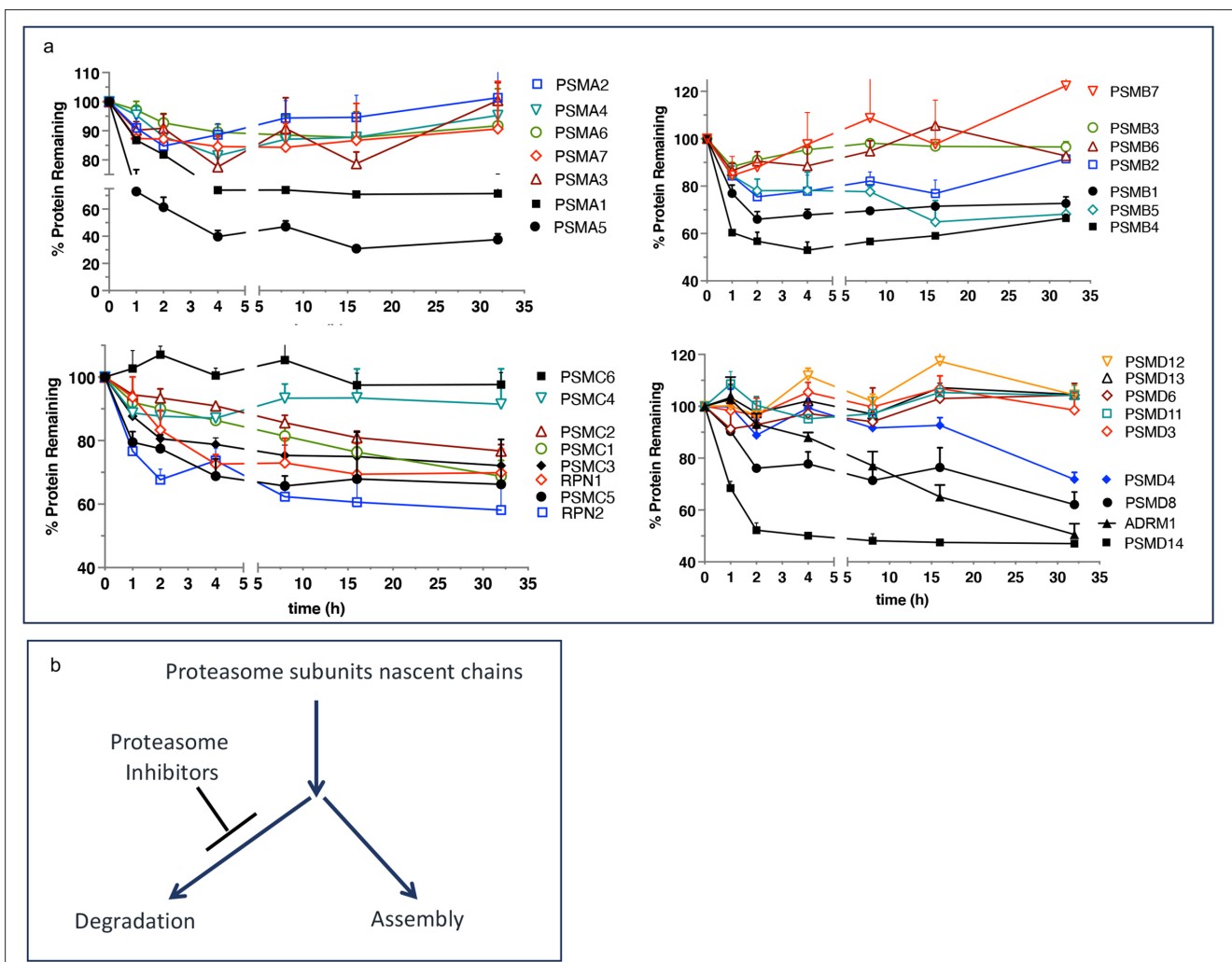

**Figure 4.** Escape from rapid degradation of nascent subunits can explain rapid recovery of proteasome activity. (**a**) Turnover of proteasome subunit in human RPE-1 cells was measured by quantitative mass-spectrometry following 1 hr labeling with heavy isotopes. Data taken from Table S4 in *McShane et al., 2016*; n=2-3. (**b**) Proposed model of how nascent proteasome subunits are partitioned between assembly and degradation.

The online version of this article includes the following source data for figure 4:

**Source data 1.** Prism file containing data from Table S4 in *McShane et al., 2016* that was used to create figure.

incorporation into the mature proteasomes is not known. One study found that proteasomes degrade a significant fraction of nascent proteasome subunits within 2–4 hr after synthesis (*McShane et al., 2016*), after which the remaining fraction is highly stable (*Figure 4a*). We hypothesize that nascent proteasome subunits are partitioned between immediate degradation and assembly, and the inhibition of the proteasome blocks degradation and increases the efficiency of proteasome assembly (*Figure 4b*). Increased expression of proteasome assembly chaperone POMP and an increase in proteasome assembly intermediates after treatment with PIs has been previously reported see Figure 6 in *Meiners et al., 2003*. If nascent polypeptides take 1–2 hr to assemble into proteasomes, this model explains translation-independent recovery of proteasome activity in the first hour after the removal of PIs (*Figure 3a*). The fraction of nascent polypeptides degraded may be much larger than in *Figure 4* because that experiment used 1 hr pulse labeling and was, therefore, unable to detect nascent proteins that are degraded within minutes after synthesis. Thus, partitioning proteasome nascent polypeptides between degradation and assembly allows cells to instantaneously upregulate proteasome biogenesis immediately after proteasome inhibition. This model will be tested in future experiments.

# Materials and methods

## Key resources table

| Reagent type (species) or resource | Designation | Source or reference | Identifiers | Additional information |
|---|---|---|---|---|
| Cell line (*Homo sapiens*) | HAP1-wt | Horizon Discovery | RRID:CVCL_Y019, Cat # C631 | Parenteral cell line (clone 631) for DDI2 KO cells below. https://horizondiscovery.com/en/engineered-cell-lines/products/hap1-parental-cell-lines |
| Cell line (*Homo sapiens*) | HAP1-DDI2 KO, clone 010 | Horizon Discovery | Cat # HZGHC000396c010 | Generated by CRISPR using gRNA:AATAGCTATGGAAGAGGCTC; 41 bp deletion; https://horizondiscovery.com/en/search?searchterm=HZGHC000396c010, |
| Cell line (*Homo sapiens*) | HAP1-DDI2 KO, clone 023 | Horizon Discovery | Calatogue # HZGHC000182c023 | Generated by CRISPR using gRNA:GCTCGAAGTCGGCGTCGACC; 1 bp insertion; https://horizondiscovery.com/en/search?searchterm=HZGHC000182c023 |
| Cell line (*Homo sapiens*) | HAP1-DDI2 KO, clone 006 | Horizon Discovery | Cat # HZGHC000182c006 | Genertaed by CRISPR using gRNA GCTCGAAGTCGGCGTCGACC; 4 bp deletion; https://horizondiscovery.com/en/search?searchterm=HZGHC000182c006 |
| Cell line (*Homo sapiens*) | MDA-MB-231 | ATCC | Cat# HTB-26 | https://www.atcc.org/products/htb-26#detailed-product-information |
| Cell line (*Homo sapiens*) | SUM149 | BioIVT | RRID:CVCL_3422 | |
| Cell line (*Homo sapiens*) | HCT-11, wt | https://doi.org/10.7554/eLife.18357 | RRID:CVCL_0291 | A matching wt clone to a mutant below, provided by Murata laboratory |
| cell line (*Homo sapiens*) | HCT-116, DDI2--D252N | https://doi.org/10.7554/eLife.18357 | | Contains CRISPR-generated D252N mutation in the active site of DDI2, provided by Murata laboratory |
| Transfected construct (*Homo sapiens*) | DDI2 siRNA10 | Horizon Discovery - Dharmacon | J-032713-10-0050 | Sequences: GGACAUGCUUAAACGGCAC |
| Transfected construct (*Homo sapiens*) | DDI2 siRNA12 | Horizon Discovery - Dharmacon | J-032713-12-0050 | Sequence: CAAGAAAGGAUUCGUCUGU |
| Transfected construct (*Homo sapiens*) | Non-targeting pool siRNA | Horizon Discovery - Dharmacon | D-001810-10-20 | Sequences: UGGUUUACAUGUCGACUAA, UGGUUUACAUGUUGUGUGA, UGGUUUACAUGUUUUCUGA, UGGUUUACAUGUUUUCCUA |
| Antibody | Anti-TCF11/NRF1 D5B10 (rabbit mAb) | Cell Signaling | Cat# 8052 S | WB (1:500) |
| Antibody | Anti-GAPDH D4C6R (mouse mAb) | Cell Signaling | Cat# 97166 | WB (1:1000) |
| Antibody | Anti-β-actin 8H10D10 (mouse mAb) | Cell Signaling | Cat #3700 | WB (1:1000) |

*Continued on next page*

*Continued*

| Reagent type (species) or resource | Designation | Source or reference | Identifiers | Additional information |
|---|---|---|---|---|
| Antibody | Anti-DDI2 (rabbit pAb) | Bethyl Laboratories | Cat# A304-629A | WB (1:5000) |
| Antibody | Anti-rabbit IgG, HRP-linked (goat) | Cell Signaling | Cat#7074 | WB (1:1000) |
| Antibody | Anti-mouse IgG, HRP-linked (goat) | Cell Signaling | Cat#7076 P2 | WB (1:1000) |
| Antibody | Goat anti-Rabbit IgG, Alexa Fluor Plus 647 | Thermofisher - Invitrogen | Cat#A32733 | WB (1:3500) |
| Antibody | Goat anti-Rabbit IgG, Alexa Fluor 680 | Thermofisher - Invitrogen | Cat#A-21076 | WB (1:3500) |
| Antibody | IRDye 800CW Goat anti-Mouse IgG | LI-COR | Cat#926–32210 | WB (1:3500) |
| Commercial assay or kit | DharmaFECT 1 | Horizon Discovery - Dharmacon | T-2001–03 | Transfection reagent for MDA-MB-231 and SUM-149 cells |
| Commercial assay or kit | Proteasome-Glo Assay | Promega | G8622 | Assay for Chymotrypsin-like |
| Commercial assay or kit | CellTiter-Glo Assay | Promega | G7572 | Assay for Cell Viability |
| Commercial assay or kit | Pierce Coomassie Plus (Bradford) Assay | ThermoFisher - Life Technologies | 23238 | Assay for Protein Quantification |
| Commercial assay or kit | TRIzol Reagent | ThermoFisher - Life Technologies | 15596018 | RNA Isolation |
| Commercial assay or kit | High-Capacity cDNA Reverse Transcription kit | Thermofisher - Applied Biosystems | 4368814 | |
| Commercial assay or kit | 2 x SYBR Green Bimake qPCR Master Mix | Selleckchem - Bimake | B21203 | |
| Commercial assay or kit | RNasin Plus Ribonuclease Inhibitor | Promega | N2615 | |
| Chemical compound, drug | Bortezomib | LC Laboratories | AS# 179324-69-7, Cat# B-1408 | Proteasome Inhibitor, |
| Chemical compound, drug | Carfilzomib | LC Laboratories | CAS# 868540-17-4, Cat# C-3022 | Proteasome Inhibitors, |
| Chemical compound, drug | CB-5083 | Cayman Chemicals | CAS# 1542705-92-9, Cat# 19311 | p97 inhibitor, |
| Chemical compound, drug | CHAPS (3-((3-cholamidopropyl) dimethylammonio)–1-propanesulfonate) | Thermo Scientific | CAS# 331717-45-4, Cat # 28300 | Detergent |
| Chemical compound, drug | Cycloheximide | Sigma-Aldrich | CAS# 66-81-9, Cat #C1988 | Protein Synthesis Inhibitor, |
| Chemical compound, drug | Digitonin | GoldBio | CAS# 11024-24-1, Cat# D-180–250 | Detergent |
| Chemical compound, drug | PhosSTOP | Roche | Cat# 4906837001 | Mixture of Phosphatase Inhibitors |
| Chemical compound, drug | Suc-LLVY-AMC | Bachem | CAS# 94367-21-2, Cat # 4011369 | Proteasome substrate |
| Chemical compound, drug | Resazurin sodium salt | Sigma-Aldrich | CAS# 62758-13-8, Cat#R7017 | Alamar Blue Viability Assay |
| Software, algorithm | PRISM | GraphPad | | version 10 |

## Source of materials

HAP1 cells (wt-clone 631, DDI2 KO clones 006, 023, and 010, Key resources table) were obtained from Horizon Discovery. MDA-MB-231 cells were purchased from ATCC (Cat. #HTB-26), and SUM149 cells

(CVCL_3422) were obtained from Asterand (*Weyburne et al., 2017*). A CRISPR-generated clone of HCT-116 cells, in which catalytic Asp-252 residue of the DDI2 gene was mutated into an asparagine (D252N) (*Koizumi et al., 2016*), and a control clone carrying wt-DDI2 allele were kindly provided by Dr. Shigeo Murata, and tested negative for Mycoplasma contamination. All cell lines were authenticated by STR profiling. Sources of inhibitors and other chemicals are listed in the Key resources table.

## Cell culture

All cells were cultured at 37 °C in a humidified atmosphere with 5% $CO_2$. HAP1 cells were cultured in Iscove's medium supplemented with 10% Fetal Bovine Serum (FBS). MDA-MB-231 and SUM149 cells were cultured in Dulbecco's Modified Eagle's Medium (DMEM)/Hams F-12 50/50 Mix supplemented with 5% FBS. SUM149 cell media were also supplemented with 4.8 µg/mL insulin, 10 mM HEPES, pH 7.3, and 1 µg/mL hydrocortisone. HCT-116 cells were cultured in McCoy medium supplemented with 10% FBS. In addition, all media were supplemented with 100 µg/mL Penicillin-streptomycin, 0.2 µg/mL ciprofloxacin (to prevent Mycoplasma contamination), and 0.25 µg/mL amphotericin B. Cells were plated overnight before treatment, then treated with inhibitors for 1 hr in a fresh medium. The inhibitor-containing medium was aspirated, except for the experiments in *Figure 1—figure supplement 2a*, where it was shaken off, and the cells were cultured in a drug-free medium for times indicated when they were harvested and analyzed as described in the figure captions. siRNAs were transfected 72 hr before treatments. The MDA-MB-231 or SUM149 cells were seeded in six-well plates at $2 \times 10^5$ cells/well the day before the transfections. The cells were transfected with 25 nM DDI2 siRNAs by using 0.3% DharmaFECT 1 in Gibco Opti-MEM 1 X Reduced Serum Medium and Corning DMEM:F-12(1:1) without antibiotics and amphotericin B. Cell viability was assayed with CellTiter-Glo (Promega) or Alamar Blue (resazurin).

## Proteasome activity assays

The activity of the proteasome's β5 sites was determined either by Succinyl(Suc)-LLVY-AMC (7-amido-4-methylcoumarin) fluorogenic substrate or by the Proteasome-Glo assay (Promega), a luciferase coupled assay, which uses Suc-LLVY-aminoluciferin as a substrate (*Britton et al., 2009*; *Moravec et al., 2009*). In the Proteasome-Glo assays, the cells in 96-well plates were washed with PBS and lysed by one cycle of freezing and thawing in 25 µL of cold PBS containing 0.05% digitonin. 25 µL of Suc-LLVY-aminoluciferin containing Proteasome-Glo reagent was added, and plates were preincubated on a shaker for ~10 min at room temperature before luminescence measuring using a mixture of PBS and Proteasome-Glo reagent as a blank. Each sample contained three technical replicates.

To determine proteasome's β5 activity in the cell extracts, cells were lysed in ice-cold 50 mM Tris-HCl, pH 7.5, 10% glycerol, 0.5% CHAPS, 5 mM MgCl2, 1 mM EDTA, 100 µM ATP, 1 mM DTT, and 1 x PhosSTOP. The cells were incubated for 15 min on ice, centrifuged at 20,000 × *g* for 15–20 min at 4 °C, and the supernatants were used for the experiments. Protein concentrations were determined using Pierce Coomassie Plus (Bradford) Assay reagent (Cat. #23238) with bovine serum albumin as a standard. An aliquot of cell lysate containing 1 µg of protein was spiked into a 100 µL per well of the 26 S assay buffer (50 mM Tris-HCl pH 7.5, 40 mM KCl, 2 mM EDTA, 1 mM DTT, and 100 µM ATP) containing 100 µM of Suc-LLVY-AMC. The mixture was thoroughly mixed and preincubated at 37 °C for 10 min. An increase in fluorescence was monitored continuously at 37 °C at the excitation wavelength of 380 nm and emission of 460 nm. The slopes of the reaction progress curves for three technical replicates were averaged, and the inhibition was calculated as a percentage by dividing the slopes of the inhibitor-treated samples by the slope of mock-treated controls. Assays were calibrated with AMC standard (*Kisselev and Goldberg, 2005*).

## Western blotting

Lysates were prepared, and total protein was quantified as described above for the fluorescent proteasome assays. Samples were mixed with lithium dodecyl sulfate loading buffer and heated before fractionation on either NuPAGE Bis-Tris 8% Midi Gel (Invitrogen, Cat. #WG1003BOX) or SurePAGE Bis-Tris 8% mini gel (GenScript, Cat. #M00662), using MES-SDS running buffer (GenScript Cat. #M00677). The proteins were transferred on 0.2 µm pore-diameter Immobilon–pSQ PVDF membrane (Cat. #ISEQ00010) using Invitrogen Power Blotter 1-Step Transfer Buffer (Thermo Cat. #PB7300). The

membrane was blocked with 5% Milk in TBST and probed with antibodies listed in the Key resources table.

## RNA isolation and qPCR

The mRNA was isolated from cells using TRIzol Reagent (Thermo Fisher Scientific Cat. #15596018) according to the manufacturer's protocol. Then, cDNA synthesis was performed using a High-Capacity cDNA Reverse Transcription kit (Applied Biosystems cat. #4368814). Before the qPCR run, the RNA and cDNA were quantified by UV absorbance using NanoDrop2000 (Thermo Scientific). The Real-time qPCR was performed using 2 x SYBR Green Bimake qPCR Master Mix on a Bio-Rad C1000 thermal cycler CFX96 Real-Time System. The primers are listed in *Supplementary file 1*.

Polysome profiling was conducted according to a published procedure (*He and Green, 2013*; *Morita et al., 2013*). Cells were washed in a cold PBS containing 100 µg/mL CHX and were resuspended in the hypotonic buffer containing 5 mM Tris-HCl pH 7.5, 2.5 mM $MgCl_2$, 1.5 mM KCl, 1 x Complete protease inhibitor (EDTA-free), 100 µg/mL CHX, 1 mM DTT, and 0.2 units/mL RNAsin Plus. Triton X-100 and sodium deoxycholate were added to the cell suspension to a final concentration of 0.5%, followed by centrifugation at 20,000 g for 15–20 min 4 °C. Extracts were loaded on 5–55% gradients of sucrose in 20 mM HEPES-KOH, pH 7.6, 0.1 M KCl, 5 mM $MgCl_2$, 100 µg/mL cycloheximide, 1 x complete EDTA free protease inhibitor cocktail, and 100 units/mL RNAsin. Following the centrifugation at 35,000 rpm for 2.5 hr at 4 °C, the gradients were manually fractionated into 200 µL fractions. Fractions containing the non-translated mRNA, 80 S ribosomes, and polysomes were pooled. The RNA was isolated and quantified by quantitative RT-PCR.

## Statistical analysis

Data points on all figures are averages +/-S.E.M. of n biological replicates, and n is provided in figure captions. Statistical analysis was carried out in GraphPad PRISM and used mixed-effect multiple comparisons on *Figure 1f* and a t-test on *Figure 2b*. p-values <0.05 were considered significant.

## Acknowledgements

This work was supported by a 5R01CA213223 grant from the NCI to AFK. Ibtisam was supported by the LPDP scholarship from the Indonesia Endowment Fund for Education. The authors are grateful to Addison Wilson for generating data for a panel b in the Supplement 2 to *Figure 1*, to Dr. Shigeo Murata for providing CRISPR-engineered HCT-116 cells that express the D252N-DDI2 mutant, to Dr. Zhe Sha, and late Dr. Alfred L Goldberg for providing Q-PCR primers, to Dr. Wade Harper and Dr. Alfred L Goldberg for advice, and to Dr. Tyler Jenkins and Sriraja Srinivasa for the critical reading of the manuscript.

## Additional information

### Competing interests

Alexei F Kisselev: AFK is a founder and Chief Scientific Officer of InhiProt LLC. The other author declares that no competing interests exist.

### Funding

| Funder | Grant reference number | Author |
| --- | --- | --- |
| National Cancer Institute | 5R01CA213223 | Alexei F Kisselev |
| Indonesia Endowment Fund for Education | | Ibtisam Ibtisam |

| Funder | Grant reference number | Author |
|--------|------------------------|--------|

The funders had no role in study design, data collection and interpretation, or the decision to submit the work for publication.

## Author contributions
Ibtisam Ibtisam, Conceptualization, Investigation, Methodology, Writing – review and editing; Alexei F Kisselev, Conceptualization, Supervision, Funding acquisition, Investigation, Methodology, Writing – original draft, Project administration, Writing – review and editing

## Author ORCIDs
Ibtisam Ibtisam (ID) https://orcid.org/0000-0002-3897-6936
Alexei F Kisselev (ID) https://orcid.org/0000-0002-6503-4995

Reviewer #1 (Public review): https://doi.org/10.7554/eLife.91678.3.sa1
Reviewer #2 (Public review): https://doi.org/10.7554/eLife.91678.3.sa2
Author response https://doi.org/10.7554/eLife.91678.3.sa3

# Additional files

## Supplementary files
• MDAR checklist
• Supplementary file 1. PCR primers used in this work.

## Data availability
Al data generated during this study are included in the manuscript and source files.

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
