## [Editor Report · eLife assessment]

The study presents **important** findings regarding a transcription-independent component of the early recovery of proteasome activity from a short pulse of proteasome inhibitor treatment, which has not been appreciated before and which is independent of the DDI2-NRF2 axis. While the evidence is in principle **solid**, with recapitulation in several cell line models, the proposed alternative underlying mechanism, namely regulation at the level of proteasome assembly, lacks experimental support, and at this point remain speculative.

---

## [Referee Report · Reviewer #1 (Public review)]

Summary:

There has been substantial prior work trying to understand the transcriptional control of proteasome expression as an adaptive response to proteasome inhibition. This field has been mired by fierce debates over the role of the protease Ddi2 in activating the transcription factor Nrf1/NFE2L1. As the authors of this manuscript point out, most of the previous research centers on the continuous treatment of cells with proteasome inhibitors rather than a brief pulse of inhibition that better models the situation when these drugs are used clinically. The authors find that the initial recovery of proteasome activity is independent of Ddi2 and involves a mechanism distinct from transcription. The authors intriguingly point to a model in which the assembly of proteasomes is regulated. If true, this would be a significant finding, but for now, this model remains more speculative.

Strengths:

The pulsed treatment of proteasome inhibitors is a strength of this lab that few others use. It better mimics the clinical use of these inhibitors and allows for a more detailed analysis of the initial response to inhibition. The authors have used multiple different clones of Ddi2 knockouts and siRNA against Ddi2 to rule out the necessity of Ddi2 in the early production of proteasomes when cells are inhibited with proteasomes. establishing a thorough knockout approach while also avoiding compensatory mutations. These experiments are well controlled, showing both the levels of Ddi2 upon knockout or knockdown and demonstrating that the cleavage of Nrf1, one of two known targets of Ddi2, is impaired. However, it should be noted that faint bands for Ddi2 mysteriously remain even in the knockout.

This article sensitively monitors the recovery of proteasome function with the β5 activity assay and for the production of new proteasome transcripts by qPCR. This precision and a detailed analysis of the timing are strengths that pointed to a more rapid recovery than transcription alone.

Weaknesses:

This paper's major weakness is the difficulty in establishing the authors' model that assembly is regulating this process. They do a convincing job demonstrating that activity recovers before transcription. The evidence that translation is unaffected depends entirely on the polysome RNA profiling from two replicates. Clearer and orthogonal data would help establish this finding. The stability of subunits is interesting and important in its own right.

In short, the authors establish that Ddi2 is unnecessary for the initial, non-transcriptional recovery of proteasome activity after a pulse of proteasome inhibition.

It is not clear what clinical impact this work will have. Although it models the pulse of proteasome inhibition more perfectly, it only looks at a single pulse rather than multiple treatments. Thus, ruling out Ddi2's importance for clinical benefit may be premature. More significantly, this work suggests that assembling proteasomes might be a regulated process worth substantial follow-up that will be interesting to follow.

---

## [Referee Report · Reviewer #2 (Public review)]

Summary:

In this work Ibtisam and Kisselev explore the role of DDI2 in the proteasome function recovery after a clinically relevant pulse dosing using different proteasome inhibitors and their corresponding PK properties. The authors report that despite lack of NRF1 activation by DDI2 there was no difference in recovery from pulsed proteasome inhibition observed in DDI2 KO cells as compared to WT controls suggesting DDI2 is not required for recovery in this system. They further show that transcription of the proteasome subunits is initiated only after partial recovery of proteasome activity is already observed suggesting that non-transcriptional mechanisms might be also involved. The authors further show that translation inhibition blocked the recovery from proteasome inhibitors.

Strengths:

Overall, it is very important and informative to use a pulse treatment type approach (mimicking the PK properties of the drugs) to explore the biology of PIs as used in this study. The authors also provide convincing data that DDI2 is not required for proteasome activity recovery post-PI pulse treatment in the systems they explored.

Weaknesses:

The authors show that the recovery of one specific catalytic activity of the proteasome post-PI treatment is transcription independent. However, in this work they do not explore the other catalytic activities of the proteasome, the protein levels of the individual subunits and most importantly the level of the different assembled proteasome complexes and how they change over time. Without this data the proposed mechanism is still speculative, in particular the conclusion on the role of translation, and ignores other findings in the field that suggest that alternative mechanisms (such as proteasome complex assembly regulation for instance) might be just as plausible.

---

## [Author Response]

The following is the authors’ response to the original reviews.

We are grateful to the reviewers for recognizing the importance of our work on transcription-independent early recovery of proteasome activity. We also thank them for their thoughtful criticisms and suggested improvements, which we addressed in the revised version as described below.

The reviewers and editors asked for data to support the model that early recovery of proteasome activity is due to accelerated proteasome assembly. This model is backed by published data that proteasome assembly intermediates increase dramatically in cells treated with proteasome inhibitors (Fig. 6 in Ref. 46 of the revised manuscript). We expanded the discussion of this paper in a paragraph that describes our model. Another key experiment to confirm this model would be to determine what fraction of nascent polypeptides is degraded within minutes after synthesis, which is not trivial, and Ibtisam ran out of time to conduct these experiments because she had to graduate in spring before the expiration of her visa. This type of experiment usually uses metabolic labeling by a heavy or radioactive amino acid that always includes a prior depletion of a non-labeled amino acid. However, the fundamental flaw of this approach, which is not recognized by the scientific community, is that depletion of an amino acid stresses cells and reduces the rate of protein synthesis, especially if this amino acid is methionine. Thus, this model is not easy to test, and should be considered a speculation. We therefore moved the description of this model, together with Fig. 4, into a separate "Ideas and Speculations" section and removed this model's description from the abstract.

Reviewer 1 raised the possibility that a background band detected on the western blot of DDI2 KO cells could be a highly homologous protease DDI1. This is highly unlikely because, according to Protein Atlas, DDI1 is selectively expressed in the testis and is not expressed in the cell lines we used. Reviewer 1 also suggested that we should base our conclusion on Nrf1 KD, which we de-facto did because we confirmed that DDI2 KD blocks Nrf1 activation (Fig. 1d).

In response to Reviewer 1 critiques regarding the presentation of proteasome subunits stability data in Fig. 4 (Ref. 45 of the revised manusript), we removed PSMB8 and replaced chaperons with the subunits of the 26S base. We changed color palettes, symbols, and axis scales to improve clarity.

We acknowledged in the discussion that our work did not exclude DDI2 role in the recovery of proteasome after repeated pulse treatments, as suggested by Reviewer 1.

We agree with Reviewer 2 that using “proteasome levels” is inaccurate when describing our activity measurement data. However, in the manuscript, we use "levels" only when discussing data in the literature. We believe measuring activity and not the total levels is more important because not all proteasomes are active, e.g., latent 20S proteasome core particles.

Reviewer 3 expressed concern that our conclusions were based on data in HAP1 cells, which are haploid, and appear not very sensitive to proteasome inhibitors. This is why we used DDI2 KD in MDA-MB-231 and SUM149 cells, which are highly sensitive to proteasome inhibitors (Weyburne et al., Ref. 11). In our experience, full extent of proteasome inhibitor cytotoxicity is not revealed until 48hr after treatments, and viability determined at 12hr and 24hr as on Fig. 1c should not be used to determine sensitivity (it was used for activity assay normalization). We added a new supplementary figure showing that HAP1 cells are as sensitive to proteasome inhibitors as MDA-MD-231 cells when cell viability is assayed 48hr after treatment (new Fig. S2). Another panel on this new figure demonstrates that the baseline proteasome activity is very similar in HAP1, MD-MB-231 and SUM149 cells. We also added data demonstrating that inactivation of DDI2 by mutation does not change the recovery of proteasome activity in HCT-116 cells (new Fig. 1g). Recovery in MDA-MB-231, SUM149, and HCT-116 cells was measured at 18hr, which is still within the 12 – 24hr window when other investigators observed partially DDI2-dependent recovery.

We have conducted an experiment in which we followed activity recovery for up to 72hr. We found that activity plateaued at 24hr and opted against the repeat because there were no changes. We feel that the manuscript should not include one biological replicate data. The fact that the recovery is incomplete and that cells seem to survive with lower levels of proteasome activity is interesting; however, investigating the molecular basis for this phenomenon is beyond the scope of the current project.

We were not disputing the conclusions of previous studies that DDI2/Nrf1 is responsible for enhanced expression of proteasomal mRNA in cells continuously treated with proteasome inhibitors. In fact, we confirmed that pulse-treatment causes similar increase (Fig. 2b). As for papers that measured activity recovery after pulse treatment, we objectively discuss our results in the context of these papers. In response to Reviewers' recommendations and minor points:

We reviewed the revised version carefully to eliminate spelling and grammatical errors and typos.We no longer refer to DDI2 as a novel protease, as suggested by Reviewer 1.We agree with Reviewer 2 that our CHX results do not necessarily mean that recovery involves translation of proteasomal mRNAs, and we now conclude that proteasome recovery requires protein synthesis.We revised Fig. 1c, 3a and 4a to improve clarity.We have stated in the caption that data in Fig. 4a comes from Table S4 in Ref. 45.We accepted an excellent suggestion of Reviewer 3 to change "recovery" to "early recovery" in the title.Regarding Reviewer 3 request to assay activity recovery at additional time points before 12h, this was done in the cycloheximide experiment in Fig. 3A.Even if we assume that the differences in the observed recovery activity in MDA-MB-231 cells (Fig. 1f) are statistically significant, which may implicate DDI2 involvement in the activity recovery, the percentage is still small, suggesting that most activity recovery is DDI2-independent.We toned down the statement "the present findings suggest that DDI2 desensitizes cells to PI by a different mechanism," replacing "suggest" with "raise a possibility".We indicated that only Bortezomib is approved for mantle cell lymphoma.We changed the description of clinical dosing as suggested by Reviewer 3. We added a reference on PK of subcutaneous bortezomib (Ref. 9), even though the review we quoted (Ref. 7) discussed subcutaneous dosing.